# Renal Replacement Therapy in Patients with Influenza Pneumonia Related Acute Respiratory Distress Syndrome

**DOI:** 10.3390/jcm10091837

**Published:** 2021-04-23

**Authors:** Ko-Wei Chang, Shaw-Woei Leu, Shih-Wei Lin, Shinn-Jye Liang, Kuang-Yao Yang, Ming-Cheng Chan, Wei-Chih Chen, Han-Chung Hu, Wen-Feng Fang, Yu-Mu Chen, Chau-Chyun Sheu, Ming-Ju Tsai, Hao-Chien Wang, Ying-Chun Chien, Chung-Kan Peng, Chieh-Liang Wu, Kuo-Chin Kao

**Affiliations:** 1Department of Thoracic Medicine, Chang Gung Memorial Hospital, Taoyuan 333, Taiwan; b9302072@cgmh.org.tw (K.-W.C.); swleu@cgmh.org.tw (S.-W.L.); ec108146@cloud.cgmh.org.tw (S.-W.L.); h3226@cgmh.org.tw (H.-C.H.); 2Division of Pulmonary and Critical Care, Department of Internal Medicine, China Medical University Hospital, Taichung 404, Taiwan; reagon6142@gmail.com; 3Department of Chest Medicine, Taipei Veterans General Hospital, Taipei 122, Taiwan; kyyang@vghtpe.gov.tw (K.-Y.Y.); wiji.chen@gmail.com (W.-C.C.); 4Institute of Emergency and Critical Care Medicine, School of Medicine, National Yang-Ming University, Taipei 122, Taiwan; 5Division of Critical Care and Respiratory Therapy, Department of Internal Medicine, Taichung Veterans General Hospital, Taichung 407, Taiwan; mingcheng.chan@gmail.com; 6College of Science, Tunghai University, Taichung 407, Taiwan; 7Department of Respiratory Therapy, College of Medicine, Chang-Gung University, Taoyuan 333, Taiwan; 8Division of Pulmonary and Critical Care Medicine, Department of Internal Medicine, Kaohsiung Chang Gung Memorial Hospital, Kaohsiung 833, Taiwan; wenfengfang@yahoo.com.tw (W.-F.F.); blackie@adm.cgmh.org.tw (Y.-M.C.); 9Department of Respiratory Care, Chang Gung University of Science and Technology, Chiayi 613, Taiwan; 10Division of Pulmonary and Critical Care Medicine, Kaohsiung Medical University Hospital, Kaohsiung 807, Taiwan; sheu@kmu.edu.tw (C.-C.S.); siegfriedtsai@gmail.com (M.-J.T.); 11Department of Internal Medicine, School of Medicine, College of Medicine, Kaohsiung Medical University, Kaohsiung 807, Taiwan; 12Division of Chest Medicine, Department of Internal Medicine, National Taiwan University Hospital, Taipei 100, Taiwan; haochienwang@gmail.com (H.-C.W.); firstaidg@gmail.com (Y.-C.C.); 13Division of Pulmonary and Critical Care Medicine, Department of Internal Medicine, Tri-Service General Hospital, National Defense Medical Center, Taipei 114, Taiwan; kanpeng@mail.ndmctsgh.edu.tw; 14Center for Quality Management, Taichung Veterans General Hospital, Taichung 407, Taiwan; cljeff.wu@gmail.com; 15Office of Medical Administration, Taichung Veterans General Hospital, Taichung 407, Taiwan

**Keywords:** renal replacement therapy, influenza, acute respiratory distress syndrome

## Abstract

Acute kidney injury (AKI) requiring renal replacement therapy (RRT) increases the mortality of acute respiratory distress syndrome (ARDS) patients. The aim of this study was to investigate the outcomes and predictors of RRT in patients with influenza pneumonia-related ARDS. This retrospective cohort study includes patients from eight tertiary referral centers in Taiwan between January and March 2016, and all 282 patients with influenza pneumonia-related ARDS were enrolled. Thirty-four patients suffered from AKI requiring RRT, while 16 patients had underlying end stage renal disease (ESRD). The 30- and 60-day mortality rates were significantly higher in patients with AKI requiring RRT compared with those not requiring RRT (50.0% vs. 19.8%, *p* value < 0.001; 58.8% vs. 27.2%, *p* value = 0.001, respectively), but the patients with ESRD had no significant difference in mortality (12.5% vs. 19.8%, *p* value = 0.744; 31.3% vs. 27.2%, *p* value = 0.773, respectively). The predictors for AKI requiring RRT included underlying chronic liver disease and C-reactive protein. The mortality predictors for patients with AKI requiring RRT included the pneumonia severity index, tidal volume, and continuous renal replacement therapy. In this study, patients with influenza pneumonia-related ARDS with AKI requiring RRT had significantly higher mortality compared with other patients.

## 1. Introduction

Acute kidney injury (AKI), defined by the Acute Kidney Injury Network [1], happened in 36–67% critically ill patients, and 5–6% of them need renal replacement therapy (RRT) [2]. For the critically ill patients who suffer from AKI and need RRT, the mortality rate is as high as 60% [3]. For patients with acute respiratory distress syndrome (ARDS), inadequate mechanical ventilator settings might contribute to adverse renal effects [4,5].A previous study showed that the incidence of AKI in patients with ARDS was 44.3% and the mortality rate was up to 42.3% [6]. Another study which investigated patients with prolonged mechanical ventilation revealed that the weaning and survival rates were lower in patients with RRT which started in the ICU compared with those with end stage renal disease (ESRD) [7].

Severe influenza infection is one of the most common etiologies of ARDS [8,9]. In the H1N1 influenza pandemic in 2009, 49–72% of influenza pneumonia patients who were admitted to the ICU had ARDS [10,11]. In the first quarter of 2016, 1,735 patients were admitted to the ICU with complicated influenza infections in Taiwan [12].

Previous studies have evaluated the outcomes of influenza induced ARDS patients with AKI [13,14], but the studies for the influence of RRT is scanty. We conducted a multicenter, retrospective study of patients admitted to ICUs due to influenza with ARDS.

## 2. Materials and Methods

### 2.1. Study Patients and Data Collection

This retrospective cohort study was conducted by the Taiwan Severe Influenza Research Consortium which includes eight tertiary referral centers in Taiwan. All patients who were admitted to the ICU at these 8 centers with virology-proven influenza between January and March in 2016 were enrolled in the current study. The patient’s demographic data, laboratory data, ventilator parameters, therapy process, and outcomes were obtained from their electronic medical records using a standard case report form. ARDS was diagnosed according to the Berlin criteria [15].

Chronic liver disease was considered as liver cirrhosis or chronic viral hepatitis, and chronic kidney disease was defined as a baseline estimated glomerular filtration rate of <60. The dynamic driving pressure was defined as peak inspiratory pressure minus positive end expiratory pressure, and compliance was defined as the tidal volume divided by the dynamic driving pressure. The patient’s laboratory data, arterial blood gas, ventilator settings, and severity scores—including the Pneumonia Severity Index (PSI) [16], Acute Physiology and Chronic Health Evaluation II (APACHE II) [17], CURB-65 pneumonia severity score [18], and Sequential Organ Failure Assessment (SOFA) score [19]—were collected on the ICU admission day. The local Institutional Review Boards for Human Research at all of the involved hospitals approved the current study. Due to the retrospective nature of the study, the need for informed consent was waived.

### 2.2. Diagnosis of Influenza

All of the patients were diagnosed with influenza by at least 1 of the following laboratory tests: nasopharynx swab or throat swab influenza rapid antigen test; nucleic acid reverse-transcriptase polymerase chain reaction, viral culture from a nasopharynx swab, throat swab, sputum or bronchoalveolar lavage, or serum antibody serologic test (antibody titers decreased >4 times from the acute to convalescent stage).

### 2.3. Renal Replacement Therapy

Indications for RRT included oliguria with fluid overload, refractory metabolic acidosis or refractory hyperkalemia. The nephrologists were consulted before the RRT was started to evaluate the indication and any contraindications, and the final decision of whether to start RRT was made by the intensive care doctors and the nephrologists together. Continuous renal replacement therapy (CRRT) was used for patients with severe shock that could not tolerate intermittent hemodialysis. CRRT could revert to intermittent hemodialysis if the patient’s blood pressure stabilized.

Every patient had a Foley catheter to monitor the urine output and avoid post-renal etiology of renal failure. Nephrotoxic antibiotics, such as aminoglycoside or colistin, were not the first-line antibiotics in our ICUs.

### 2.4. Statistical Analyses

Statistical analyses and database management were performed using SPSS version 22.0.0 (SPSS Inc., Chicago, IL, USA). We used number (percentages) for nominal variables and the mean ± standard deviation for continuous variables. Pearson’s chi-squared test was used to compare nominal variables. An independent Student’s *t*-test was used to compare two groups of continuous variables, and one-way analysis of variance was used to compare multiple groups of continuous variables. We did the post hoc test by Tukey’s honestly significant difference test. We also calculated the statistical power. Univariate and multivariate binary logistic regression were used to analyze the predictive factors for AKI requiring hemodialysis. Univariate and multivariate Cox regression were used to analyze the predictive factors of survival. Variables with *p* value less than 0.05 in univariate analysis were included for multivariate analysis. In the current study, two-tailed tests were used and statistical significance was defined as a *p* value < 0.05.

## 3. Results

### 3.1. General Data

A total of 336 patients were diagnosed with influenza and admitted to the ICU between January and March 2016, and 282 of these patients met the criteria for ARDS (Figure 1). There were 16 patients who had underlying ESRD disease on regular hemodialysis, and then received RRT during the ARDS course. There were 34 patients who suffered from AKI and received RRT which started in the ICU, and 10 of these patients had previously chronic kidney disease without hemodialysis before their hospital admission. On average, these patients started hemodialysis 3.58 ± 3.53 days after their respiratory failure began, and 16 patients received CRRT.

The patients’ demographic data, laboratory data, and ventilator parameters are shown in Table 1. The patients with AKI requiring RRT were younger than the no RRT patients and the ESRD patients, but these differences were not statistically significant. The patients with AKI requiring RRT had a higher ratio of underlying chronic liver disease or chronic kidney disease compared with the other patients, whereas the ESRD patients had a higher ratio of underlying diabetes mellitus and hypertension. The AKI patients requiring RRT and ESRD patients had significantly more severe conditions according to their PSI (AKI requiring RRT vs. no RRT: *p* value = 0.005, ESRD vs. no RRT: *p* value = 0.150), APACHE II (AKI requiring RRT vs. no RRT: *p* value =< 0.001, ESRD vs. no RRT: *p* value = 0.002), and SOFA scores (AKI requiring RRT vs. no RRT: *p* value =< 0.001, ESRD vs. no RRT: *p* value = 0.001), and they had significantly higher lactate (AKI requiring RRT vs. no RRT: *p* value = 0.003) and total bilirubin levels (AKI requiring RRT vs. no RRT: *p* value = 0.023) than no RRT patients. More patients with AKI requiring RRT needed vasopressor agents (AKI requiring RRT vs. no RRT: *p* value = 0.008, ESRD vs. no RRT: *p* value = 0.129). The platelet level was lower in patients who received RRT, but no significant difference in post hoc analysis (AKI requiring RRT vs. no RRT: *p* value = 0.150, ESRD vs. no RRT: *p* value = 0.097). More severe metabolic acidosis was noted in the AKI patients requiring RRT, but this difference was not statistically significant. Moreover, the patients with AKI requiring RRT had a poorer PaO_2_/FiO_2_ ratio (AKI requiring RRT vs. no RRT: *p* value = 0.028), and they needed higher peak airway pressure (AKI requiring RRT vs. no RRT: *p* value = 0.020).

Seventy-six patients were met the diagnosis of AKI, but they did not need RRT during the ICU course. The comparison of the AKI patients requiring RRT and not requiring RRT are shown in Table 2. The patients with AKI requiring RRT, comparing with the patients with AKI not requiring RRT, had significant higher PSI (142.0 ± 46.2 vs. 120.6 ± 45.4, *p* value = 0.025), APACHE II score (29.5 ± 8.1 vs. 23.4 ± 7.4, *p* value < 0.001), SOFA score (13.9 ± 3.5 vs. 10.3 ± 3.8, *p* value < 0.001), C-reactive protein (19.4 ± 10.2 vs. 14.9 ± 10.3, *p* value = 0.041), peak airway pressure (31.6 ± 5.1 vs. 29.1 ± 4.7, *p* value = 0.028), and significant poorer PaO_2_/FiO_2_ ratio (82.4 ± 46.7 vs. 105.7 ± 62.0, *p* value = 0.037). More patients with AKI requiring RRT needed vasopressor agents than not requiring RRT (73.5% vs. 40.8%, *p* value = 0.002).

### 3.2. Clinical Outcomes

The clinical outcomes are showen in Table 3. The patients with AKI requiring RRT had significantly higher 30- and 60-day mortality rates compared with the no RRT patients (30 days: 50.0 vs. 19.8%, *p* value < 0.001; 60 days: 58.8 vs. 27.2%, *p* value = 0.001). The statistical power for 30- and 60-day mortality were 0.977 and 0.967, respectively. However, patients with ESRD who received RRT did not have a significantly different 30-day or 60-day mortality compared with the no RRT patients (30 days: 12.5 vs. 19.8%, *p* value = 0.744; 60 days: 31.3 vs. 27.2%, *p* value = 0.773). The Kaplan–Meier curve is shown in Figure 2. The patients who had AKI without RRT during their ICU course had significantly lower 60 day mortality rate than those patients who had AKI requiring RRT (30.3% vs. 58.8%, *p* value = 0.005). Compared with the no hemodialysis survival patients, the survival patients with AKI requiring RRT had a longer ICU stay (29.9 ± 23.3 vs. 19.8 ± 18.5 days, *p* value = 0.059) and hospital stay (58.8 ± 42.9 vs. 35.3 ± 25.9, *p* value = 0.002), whereas the survival patients with ESRD receiving RRT did not have a significantly different ICU stay (25.6 ± 22.0 vs. 19.8 ± 18.5 days, *p* value = 0.331) or hospital stay (46.8 ± 35.1 vs. 35.3 ± 25.9 days, *p* value = 0.184). Patients with AKI requiring RRT also had a significantly reduced number of ventilator free days in 30- or 60-days compared with the no RRT patients or the ESRD on RRT patients. Among the patients with AKI requiring RRT, only one patient progressed to ESRD and needed long term hemodialysis after discharge from the hospital. The average duration of RRT in the other patients was 33.7 ± 20.6 days (maximum: 60 days, minimum: 10 days). The predictors of 60-day survival in all patients are shown in Table 4. In the multivariate Cox regression analysis, the AKI requiring RRT (hazard ratio: 3.548, *p* value = 0.003) and tidal volume/predicted body weight (hazard ratio: 1.317, *p* value = 0.003) were two independent risk factors.

### 3.3. Predictive Factors for Acute Kidney Injury Requiring Renal Replacement Therapy

The predictive factors for AKI requiring RRT in influenza pneumonia induced ARDS patients are shown in Table 5. Patients with underlying ESRD and RRT were excluded from the analysis. In the univariate binary logistic regression, the PaO_2_/FiO_2_ ratio, underlying chronic liver disease, underlying chronic kidney disease, initial hemoglobin, C-reactive protein (CRP), lactate, total bilirubin, first day peak airway pressure and dynamic driving pressure were significant factors. In the multivariate analysis, only underlying chronic liver disease (odds ratio: 5.446, *p* value = 0.031) and CRP (odds ratio: 1.078, *p* value = 0.036) were significant predictive factors.

### 3.4. Survival Predictors for Patients with Acute Kidney Injury Requiring Renal Replacement Therapy

The predictors of 60-day survival in patients with AKI requiring RRT are shown in Table 6. In the univariate Cox regression test, the PSI (hazard ratio: 1.014, *p* value = 0.010), and tidal volume/predicted body weight (hazard ratio: 1.218, *p* value = 0.040) were significant predictors. We also used the factors with a *p* value < 0.100 to conduct the multivariate Cox regression test, and the results revealed that the PSI (hazard ratio: 1.037, *p* value = 0.002), tidal volume/predicted body weight (hazard ratio: 1.541, *p* value = 0.022), and CRRT (hazard ratio: 4.752, *p* value = 0.045) were independent factors for predicting the patient’s 60 day survival.

## 4. Discussion

In the current study, we found that patients with influenza pneumonia induced ARDS with AKI requiring RRT had a significantly higher mortality rate compared with patients who did not require RRT. However, patients with ESRD had a similar mortality rate to patients without RRT. The most important risk factors for patients with AKI requiring RRT included chronic liver disease and high CRP, and the most important mortality predictors in patients with AKI requiring RRT were the PSI, tidal volume and CRRT.

In our study, the patients with AKI requiring RRT had higher mortality, and they also had higher severity in general condition (higher PSI, APACHE II, and SOFA score) and in respiratory condition (poorer PaO_2_/FiO_2_ ratio). The ARDS is a systemic disease that the inadequate ventilator setting in ARDS patients with barotrauma, volutrauma, or biotrauma may induce multi-organ failure [4], and inadequate PEEP setting may result adverse renal hemodynamic effects [5]. On the other side, the acute kidney injury may induce acute pulmonary edema, electrolyte imbalance, or metabolic acidosis that may worse the respiratory condition. It seem that the higher mortality in these patients may be due to overall disease status. However, in the multivariate Cox regression analysis, we found the AKI requiring RRT was one of the independent predictors for 60-day mortality (hazard ratio: 3.548, *p* value = 0.003).

Several previous studies have discussed influenza pneumonia induced ARDS patients with AKI and RRT, but they have reported conflicting results concerning patient mortality. A study with 47 patients revealed that 19.1% of patients needed hemodialysis, and that 66.7% of these patients had a significantly higher mortality than the patients without hemodialysis [20]. Another study with 89 patients showed that the incidence of AKI requiring hemodialysis was 13.5%, and that the mortality rate was 50%, which was significantly higher than the mortality rate for patients without hemodialysis [21]. A Korean study including 221 patients, found that 33 patients (14.9%) had AKI requiring hemodialysis, and that their mortality rate was significantly increased compared with the other patient groups within the study (28.2% vs. 7.9%) [22]. On the other hand, two studies showed that hemodialysis patients did not have a significantly higher mortality rate. In one study 24.0% of patients needed hemodialysis and their mortality rate was 32.1% [13]; in the other study 23.8% of patients received hemodialysis and their mortality rate was 72% [23]. In our study, 12.1% of patients had AKI which needed RRT, which was similar to that of previous studies. The 60 day mortality rate for patients with AKI requiring hemodialysis was 58.8%; this was in the middle of the reported mortality rates from previous studies. When our outcome was compared with those studies that reported a higher mortality rate, it was clear that our study included the largest number of patients.

In the current study, the ARDS patients with ESRD on regular hemodialysis did not have a significantly different mortality rate compared with the patients without RRT. However, this result was different to previous studies in general ICU patients. A study in 2006 showed that patients with ESRD who were admitted to the ICU had a significantly higher hospital mortality rate compared with patients without ESRD (45.3% vs. 31.2%) [24]. Another study in 2015 reported that the mortality rate in ESRD patients was significantly higher than in patients without ESRD (34.2% vs. 18.0%) [25]. A third study in 2017 showed that ICU patients with ESRD had higher ICU, 28-day, and in-hospital mortality rates compared with other patients (21.1% vs. 12.0%) [26]. In the current study, we focused on patients with ARDS who typically have much more severe conditions than other ICU patients. The mortality rate of ESRD patients was similar to that of previous studies, but the mortality rate of patients without hemodialysis was much higher (27.2%). Perhaps this could indicate that in patients with more severe disease, ESRD is not the reason for the increased mortality rate.

Multivariate analysis revealed that chronic liver disease and CRP were two predictive factors for patients with AKI requiring RRT. Chronic liver disease was associated with AKI development in a previous study that focused on ICU patients [6] and another study which focused on ARDS patients [27]. CRP was also found to be related to AKI in a previous study of influenza patients [20]. Some previous studies considered that CRP is not only a marker for AKI, but also plays a pathogenic role in AKI [28].

The current study also found that the PSI, tidal volume/predicted body weight and CRRT free days were predictive factors for the mortality of AKI patients requiring RRT. The fact that low tidal volume ventilation can improve survival in ARDS patients has been confirmed by a previous study [29]. In another study by our group, we found that a first day tidal volume/predicted body weight >8 mL/kg was related to an increase in mortality [30]. Higher tidal volume/predicted body weight being associated with higher mortality was also noted in AKI patients requiring RRT. A previous study showed that CRRT was associated with an increase in morality, but severity of illness was a confounder [31]. A study of influenza ARDS patients showed that patients requiring CRRT had significantly higher mortality compared with other patients [14]. In the present study, CRRT usage increased the mortality rate. Determining the predictive factors, enables doctors to know which patients are at risk of suffering from AKI or increased mortality and can allow them to take extra precautions.

There were several limitations to the current study. Firstly, this was a retrospective study, and there are many confounders that could have influenced the results. Secondly, we only focused on patients with influenza related ARDS in this study, so whether these results can be extrapolated to ARDS due to other etiologies needs further evaluation. Thirdly, indications for RRT usage are universal, but there are no clear criteria which are used in Taiwan, especially for CRRT usage. A well-designed prospective study with strict patient selection and a standardized protocol is needed to confirm the results of the present study.

## 5. Conclusions

In the current study, we found that patients with AKI requiring RRT had a significantly higher mortality rate compared with patients not requiring RRT, but that ESRD patients had a similar mortality rate compared with non RRT patients in influenza pneumonia induced ARDS patients. CRP and chronic liver disease were two independent predictive factors for patients requiring RRT. PSI score, tidal volume, and CRRT usage were independent predictive factors of mortality for AKI patients requiring RRT.

## Figures and Tables

**Figure 1 jcm-10-01837-f001:**
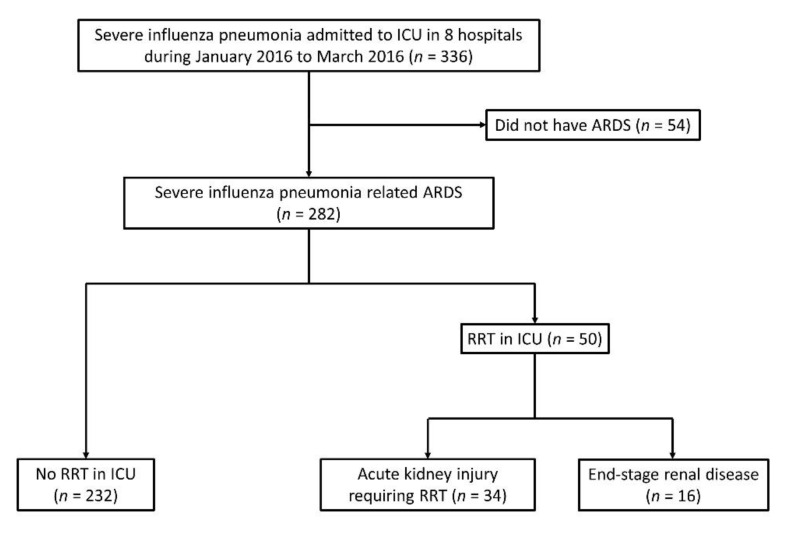
Flow chart of patients in this study; ARDS: acute respiratory distress syndrome; ICU: intensive care unit; RRT: renal replacement therapy.

**Figure 2 jcm-10-01837-f002:**
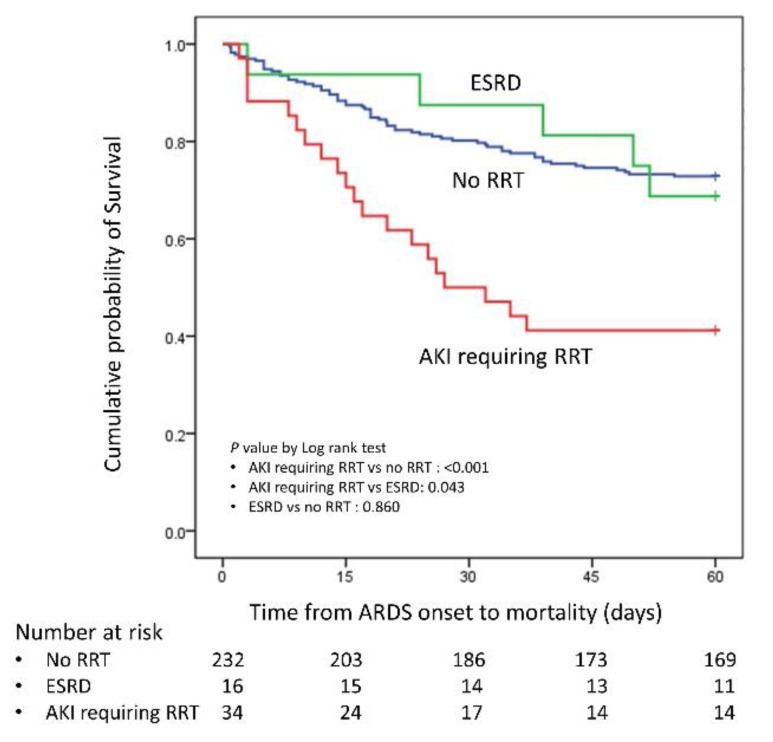
Kaplan–Meier curve for AKI requiring RRT, ESRD and no RRT; RRT: renal replacement therapy; AKI: acute kidney injury; ESRD: end stage renal disease.

**Table 1 jcm-10-01837-t001:** Demography data.

Characteristics	Total Patients	No RRT	ESRD	AKI Requiring RRT	*p* Value
	(*n* = 282)	(*n* = 232)	(*n* = 16)	(*n* = 34)	
Age (years)	60.1 ± 14.8	60.7 ± 15.2	60.4 ± 12.2	55.8 ± 12.4	0.187
Gender (male/female)	177/105	145/87	8/8	24/10	0.366
BMI (kg/m^2^)	25.2 ± 5.4	25.1 ± 5.5	23.9 ± 4.3	26.0 ± 5.0	0.438
Comorbidity					
Malignancy	39 (13.8%)	32 (13.8%)	2 (12.5%)	5 (14.7%)	0.977
Liver disease	35 (12.4%)	24 (10.3%)	1 (6.3%)	10 (29.4%)	0.005 *
Heart failure	27 (9.6%)	19 (8.2%)	4 (25.0%)	4 (11.8%)	0.078
Hypertension	118 (41.8%)	92 (39.7%)	12 (75%)	14 (41.2%)	0.021 *
Diabetes Mellitus	82 (29.1%)	60 (25.9%)	9 (56.3%)	13 (38.2%)	0.016 *
Chronic lung disease	23 (8.2%)	19 (8.2%)	1 (6.3%)	3 (8.8%)	0.952
Chronic steroid usage	17 (6.0%)	14 (6.0%)	1 (6.3%)	2 (5.9%)	0.999
Influenza type					0.170
Type A	213 (75.5%)	172 (74.1%)	12 (75%)	29 (85.3%)	
Type B	24 (8.5%)	18 (7.8%)	3 (18.8%)	3 (8.8%)	
Unknown	45 (16.0%)	42 (18.1%)	1 (6.3%)	2 (5.9%)	
Severity index					
PSI	120.1 ± 46.2	115.7 ± 44.9	137.6 ± 50.3	142.0 ± 46.2	0.002 *
CURB 65	2.3 ± 1.3	2.2 ± 1.3	2.7 ± 1.5	2.4 ± 1.0	0.379
APACHE II score	23.4 ± 8.7	22.1 ± 8.3	29.5 ± 7.3	29.5 ± 8.1	<0.001 *
SOFA score	10.6 ± 4.0	9.7 ± 3.7	14.2 ± 3.4	13.9 ± 3.5	<0.001 *
Laboratory data					
White blood cell (/uL)	10414.0 ± 6235.7	10331.9 ± 6005.3	10751.9 ± 6754.2	10810.6 ± 7582.1	0.895
Platelet (1000/uL)	161.4 ± 92.5	167.9 ± 94.1	117.2 ± 62.9	136.4 ± 85.1	0.029 *
C-reactive protein (mg/dL)	15.3 ± 10.3	14.6 ± 10.2	17.2 ± 9.9	19.4 ± 10.2	0.051
Lactate (mg/dL)	29.3 ± 36.0	26.2 ± 29.5	29.8 ± 31.3	49.8 ± 63.6	0.004 *
Albumin (mg/dL)	2.8 ± 0.6	2.9 ± 0.5	3.1 ± 0.5	2.7 ± 0.6	0.070
Total bilirubin (mg/dL)	1.0 ± 1.4	1.0 ± 1.3	0.7 ± 0.5	1.7 ± 2.0	0.019 *
Need vasopressor agents	150 (53.2%)	114 (49.1%)	11 (68.8%)	25 (73.5%)	0.013 *
Arterial blood gas					
pH	7.4 ± 0.1	7.4 ± 0.1	7.3 ± 0.2	7.3 ± 0.1	0.295
PaO_2_ (mm Hg)	100.9 ± 69.1	101.7 ± 71.0	132.9 ± 74.8	81.4 ± 46.6	0.074
PaCO_2_ (mm Hg)	42.4 ± 20.5	42.7 ± 21.3	38.2 ± 8.3	42.3 ± 18.6	0.750
HCO_3_ (mm/L)	23.2 ± 15.1	23.9 ± 16.5	20.4 ± 5.0	19.6 ± 4.5	0.281
FiO_2_	0.8 ± 0.2	0.8 ± 0.2	0.8 ± 0.2	0.9 ± 0.2	0.014 *
PaO_2_/FiO_2_ ratio (mm Hg)	107.3 ± 62.1	111.6 ± 63.8	99.7 ± 55.5	82.4 ± 46.7	0.033 *
Tidal volume/predict body weight (mL/kg)	8.5 ± 2.0	8.5 ± 1.9	8.3 ± 2.2	8.9 ± 2.4	0.523
Positive end expiratory pressure (cm H_2_O)	10.5 ± 3.8	10.5 ± 3.8	9.9 ± 5.0	11.2 ± 3.5	0.483
Peak airway pressure (cm H_2_O)	29.3 ± 4.9	28.9 ± 4.9	29.2 ± 2.7	31.6 ± 5.1	0.027 *
Dynamic driving pressure (cm H_2_O)	18.6 ± 4.9	18.3 ± 4.9	18.9 ± 4.2	20.3 ± 5.2	0.117
Compliance (mL/cm H_2_O)	28.3 ± 12.0	28.7 ± 12.7	26.0 ± 7.7	26.3 ± 9.0	0.462

** p* < 0.05 RRT: renal replacement therapy ESRD: end stage renal disease AKI: acute kidney injury PSI: pneumonia severity index APACHE II: acute physiology and chronic health evaluation II SOFA: sequential organ failure assessment.

**Table 2 jcm-10-01837-t002:** Comparison of the AKI patients requiring RRT and not requiring RRT.

Characteristics	AKI not Requiring RRT	AKI Requiring RRT	*p* Value
	(*n* = 76)	(*n* = 34)	
Age (years)	61.8 ± 16.7	55.8 ± 12.4	0.064
Gender (male/female)	47/29	24/10	0.376
BMI (kg/m^2^)	24.3 ± 5.1	26.0 ± 5.0	0.111
Comorbidity			
Malignancy	11 (14.5%)	5 (14.7%)	0.975
Liver disease	12 (15.8%)	10 (29.4%)	0.099
Heart failure	10 (13.2%)	4 (11.8%)	>0.999
Hypertension	33 (43.4%)	14 (41.2%)	0.826
Diabetes Mellitus	20 (26.3%)	13 (38.2%)	0.207
Chronic lung disease	8 (10.5%)	3 (8.8%)	>0.999
Chronic steroid usage	6 (7.9%)	2 (5.9%)	>0.999
Influenza type			0.077
Type A	50 (65.8%)	29 (85.3%)	
Type B	9 (11.8%)	3 (8.8%)	
Unknown	17 (22.4%)	2 (5.9%)	
Severity index			
PSI	120.6 ± 45.4	142.0 ± 46.2	0.025 *
CURB 65	2.5 ± 1.3	2.4 ± 1.0	0.571
APACHE II score	23.4 ± 7.4	29.5 ± 8.1	<0.001 *
SOFA score	10.3 ± 3.8	13.9 ± 3.5	<0.001 *
Laboratory data			
White blood cell (/uL)	10075.8 ± 6124.9	10810.6 ± 7582.1	0.754
Platelet (1000/uL)	156.7 ± 75.8	136.4 ± 85.1	0.113
C-reactive protein (mg/dL)	14.9 ± 10.3	19.4 ± 10.2	0.041 *
Lactate (mg/dL)	26.8 ± 23.9	49.8 ± 63.6	0.254
Albumin (mg/dL)	2.9 ± 0.5	2.7 ± 0.6	0.060
Total bilirubin (mg/dL)	0.9 ± 0.8	1.7 ± 2.0	0.369
Need vasopressor agents	31 (40.8%)	25 (73.5%)	0.002 *
Arterial blood gas			
pH	7.4 ± 0.1	7.3 ± 0.1	0.255
PaO_2_ (mm Hg)	100.6 ± 53.2	81.4 ± 46.6	0.084
PaCO_2_ (mm Hg)	43.3 ± 15.9	42.3 ± 18.6	0.781
HCO_3_ (mm/L)	22.9 ± 5.4	19.6 ± 4.5	0.007
FiO_2_	0.8 ± 0.2	0.9 ± 0.2	0.035
PaO_2_/FiO_2_ ratio (mm Hg)	105.7 ± 62.0	82.4 ± 46.7	0.037 *
Tidal volume/predict body weight (mL/kg)	8.5 ± 1.9	8.9 ± 2.4	0.363
Positive end expiratory pressure (cm H_2_O)	10.1 ± 4.2	11.2 ± 3.5	0.109
Peak airway pressure (cm H_2_O)	29.1 ± 4.7	31.6 ± 5.1	0.028 *
Dynamic driving pressure (cm H_2_O)	18.8 ± 4.8	20.3 ± 5.2	0.178
Compliance (mL/cm H_2_O)	27.3 ± 10.8	26.3 ± 9.0	0.910

** p* < 0.05 AKI: acute kidney injury RRT: renal replacement therapy PSI: pneumonia severity index APACHE II: acute physiology and chronic health evaluation II SOFA: sequential organ failure assessment.

**Table 3 jcm-10-01837-t003:** Clinical outcome.

Characteristics	No RRT	ESRD	AKI Requiring RRT	*p* Value
	(*n* = 232)	(*n* = 16)	(*n* = 34)	
Mortality—no. (%)				
At day 30	46 (19.8%)	2 (12.5%)	17 (50.0%)	<0.001 *
At day 60	63 (27.2%)	5 (31.3%)	20 (58.8%)	0.001 *
Hospital	70 (30.2%)	5 (31.3%)	20 (58.8%)	0.004 *
Length of ICU stay				
Survivors	19.8 ± 18.5	25.6 ± 22.0	29.9 ± 23.3	0.125
Non-survivors	20.1 ± 18.2	15.6 ± 12.5	13.3 ± 10.5	0.298
Length of hospital stay				
Survivors	35.3 ± 25.9	46.8 ± 35.1	58.8 ± 42.9	0.007 *
Non-survivors	26.3 ± 24.6	33.8 ± 20.3	16.8 ± 10.9	0.162
Ventilation-free days				
At day 30	11.5 ± 9.8	10.1 ± 10.4	4.1 ± 7.6	<0.001 *
At day 60	31.3 ± 22.3	28.4 ± 22.1	13.8 ± 19.4	<0.001 *

** p* < 0.05 RRT: renal replacement therapy ESRD: end stage renal disease AKI: acute kidney injury ICU: intensive care unit.

**Table 4 jcm-10-01837-t004:** Predictive factors for 60 days mortality in all patients.

	Univariate		Multivariate	
	Hazard Ratio (95% CI)	*p* Value	Hazard Ratio (95% CI)	*p* Value
Age, per 1 year increment	1.010 (0.996–1.025)	0.154		
Gender				
Female	1 (Reference)			
Male	1.115 (0.720–1.727)	0.626		
Severity score				
Pneumonia severity index, per 1 increment	1.013 (1.009–1.018)	<0.001 *	1.008 (0.999–1.017)	0.095
APACHE II, per 1 increment	1.087 (1.058–1.115)	<0.001 *	1.043 (0.986–1.103)	0.146
SOFA, per 1 increment	1.189 (1.111–1.273)	<0.001 *	1.077 (0.954–1.215)	0.231
Underlying disease				
Chronic liver disease				
No	1 (Reference)		1 (Reference)	
Yes	2.379 (1.430–3.956)	0.001 *	1.375 (0.575–3.286)	0.474
Heart failure				
No	1 (Reference)			
Yes	1.094 (0.549–2.180)	0.799		
Laboratory data				
C-reactive protein, per 1 mg/dL increment	1.012 (0.990–1.034)	0.277		
Lactate, per 1 mg/dL increment	1.013 (1.008–1.018)	<0.001 *	1.001 (0.991–1.012)	0.827
Total bilirubin, per 1 mg/dL increment	1.279 (1.126–1.454)	<0.001 *	0.919 (0.700–1.205)	0.541
Blood gas analysis and respiratory mechanism				
PaO_2_/FiO_2_ ratio, per 1 mm Hg increment	0.994 (0.990–0.998)	0.004 *	0.999 (0.993–1.006)	0.876
PaCO_2_, per 1 mm Hg increment	0.993 (0.979–1.006)	0.287		
Peak airway pressure, per 1 cm H_2_O increment	1.027 (0.980–1.077)	0.261		
Positive end expiratory pressure, per 1 cm H_2_O increment	0.943 (0.887–1.002)	0.058		
Dynamic driving pressure, per 1 cm H_2_O increment	1.058 (1.013–1.106)	0.011 *	0.951 (0.892–1.014)	0.126
Tidal volume/predicted body weight, per 1 mL/kg increment	1.232 (1.098–1.382)	<0.001 *	1.317 (1.100–1.578)	0.003 *
Intervention				
Acute kidney injury requiring renal replacement therapy				
No	1 (Reference)		1 (Reference)	
Yes	2.795 (1.695–4.610)	<0.001 *	3.548 (1.511–7.909)	0.003 *
Prone positioning				
No	1 (Reference)			
Yes	0.969 (0.588–1.595)	0.900		
Extracorporeal membrane oxygenation				
No	1 (Reference)		1 (Reference)	
Yes	2.797 (1.804–4.337)	<0.001 *	1.928 (0.870–4.273)	0.106

** p* < 0.05 APACHE II: Acute Physiology and Chronic Health Evaluation II SOFA: sequential organ failure assessment.

**Table 5 jcm-10-01837-t005:** Predictive factors for acute kidney injury requiring RRT.

	Univariate		Multivariate	
	Odds Ratio (95% CI)	*p* Value	Odds Ratio (95% CI)	*p* Value
Age, per 1 year increment	0.978 (0.955–1.002)	0.071		
PaO_2_/FiO_2_ ratio, per 1 mm Hg increment	0.990 (0.983–0.998)	0.014 *	0.990 (0.976–1.005)	1.176
Chronic liver disease				
No	1 (Reference)		1 (Reference)	
Yes	3.611 (1.543–8.450)	0.003 *	4.464 (1.235–16.130)	0.022 *
Chronic kidney disease				
No	1 (Reference)		1 (Reference)	
Yes	10.324 (3.820–27.899)	<0.001 *	9.955 (1.415–70.042)	0.021 *
Hemoglobin, per 1 g/dL increment	0.854 (0.740–0.985)	0.030 *	0.878 (0.691–1.116)	0.288
Platelet, per 1000/uL increment	0.995 (0.991–1.000)	0.061		
C-reactive protein, per 1 mg/dL increment	1.044 (1.006–1.083)	0.022 *	1.077 (1.009–1.149)	0.025 *
Lactate, per 1 mg/dL increment	1.012 (1.003–1.020)	0.005 *	1.003 (0.988–1.018)	0.684
Albumin, per 1 mg/dL increment	0.523 (0.256–1.071)	0.076		
Total bilirubin, per 1 mg/dL increment	1.266 (1.024–1.565)	0.030 *	1.343 (0.943–1.912)	0.102
Peak airway pressure, per 1 cm H_2_O increment	1.111 (1.025–1.204)	0.010 *	1.098 (0.930–1.297)	0.269
Dynamic driving pressure, per 1 cm H_2_O increment	1.082 (1.002–1.168)	0.045 *	0.991 (0.837–1.173)	0.912

** p* < 0.05 RRT: renal replacement therapy.

**Table 6 jcm-10-01837-t006:** Predictive factors for 60 days mortality in AKI requiring RRT patients.

	Univariate		Multivariate	
	Hazard Ratio (95% CI)	*p* Value	Hazard Ratio (95% CI)	*p* Value
Gender				
Female	1 (Reference)			
Male	0.608 (0.242–1.527)	0.290		
Age, per 1 year increment	1.000 (0.963–1.039)	0.986		
BMI, per 1 kg/m^2^ increment	0.981 (0.897–1.073)	0.677		
Pneumonia severity index, per 1 increment	1.014 (1.003–1.025)	0.010 *	1.061 (1.019–1.104)	0.004 *
APACHE II, per 1 increment	1.016 (0.960–1.075)	0.586		
Underlying disease				
Malignancy				
No	1 (Reference)		1 (Reference)	
Yes	2.522 (0.901–7.062)	0.078	8.153 (0.694–95.796)	0.095
Liver disease				
No	1 (Reference)			
Yes	2.106 (0.847–5.237)	0.109		
Heart failure				
No	1 (Reference)			
Yes	0.362 (0.048–2.708)	0.322		
Laboratory data				
Hemoglobin, per 1 g/dL increment	0.860 (0.724–1.022)	0.086	0.936 (0.639–1.371)	0.735
Albumin, per 1 mg/dL increment	0.761 (0.356–1.626)	0.480		
Lactate, per 1 mg/dL increment	1.006 (0.999–1.014)	0.077	1.013 (0.994–1.032)	0.175
Blood gas analysis and respiratory mechanism				
PaO_2_/FiO_2_ ratio day 0, per 1 mm Hg increment	1.003 (0.993–1.012)	0.592		
PaO_2_/FiO_2_ ratio day 1, per 1 mm Hg increment	0.999 (0.993–1.005)	0.755		
PaO_2_/FiO_2_ ratio day 2, per 1 mm Hg increment	0.998 (0.992–1.004)	0.556		
PaO_2_/FiO_2_ ratio day 7, per 1 mm Hg increment	0.994 (0.988–1.000)	0.045 *	0.994 (0.987–1.001)	0.099
PaO_2_, per 1 mm Hg increment	0.985 (0.960–1.010)	0.229		
PaCO_2_, per 1 mm Hg increment	1.004 (0.979–1.029)	0.756		
HCO_3_, per 1 mm/L increment	0.924 (0.831–1.027)	0.144		
pH, per 1 increment	0.188 (0.007–4.866)	0.314		
Positive end expiratory pressure, per 1 cm H_2_O increment	0.904 (0.797–1.027)	0.121		
Peak airway pressure, per 1 cm H_2_O increment	0.976 (0.896–1.064)	0.585		
Tidal volume/predicted body weight, per 1 mL/kg increment	1.218 (1.009–1.470)	0.040 *	1.783 (1.088–2.920)	0.022 *
Dynamic lung compliance, per 1 mL/cm H_2_O increment	1.021 (0.972–1.073)	0.400		
Intervention				
Prone positioning				
No	1 (Reference)			
Yes	1.722 (0.699–4.241)	0.237		
Extracorporeal membrane oxygenation				
No	1 (Reference)			
Yes	1.342 (0.535–3.370)	0.531		
Hemodialysis free days in 30 days, per 1 day increment	0.943 (0.841–1.056)	0.308		
CRRT free days in 30 days, per 1 day increment	0.942 (0.900–0.985)	0.008 *	0.887 (0.790–0.995)	0.041 *

** p* < 0.05 AKI: acute kidney injury RRT: renal replacement therapy CRRT: continuous renal replacement therapy.

## Data Availability

The datasets used and/or analyzed during the current study are available from the corresponding author on reasonable request.

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
