# Peer review of "Renal Replacement Therapy in Patients with Influenza Pneumonia Related Acute Respiratory Distress Syndrome"

_jcm, 2021, doi:10.3390/jcm10091837_

Round 1
Reviewer 1 Report
The reviwer thanks the authrs for their replies to the questions of all reviewers. I agree that the manuscript has seriously improved and have no further concerns.
Author Response
We thank the reviewer’s comment.
Reviewer 2 Report
The investigators present a retrospective review of AKI requiring RRT in patients with influenza pneumonia from eight tertiary referral centers in Taiwan between Jan and Mar 2016.
The study cohort consisted of 282 patient admitted with influenza that met Berlin criteria for the diagnosis of ARDS. Sixteen patients were classified as ESRD and therefore would not be considered AKI patients. The analysis is based on the remaining 34 patients, of which, 10 med criteria for CKD prior to the hospitalization (GFR < 60). 76 patients met criteria for AKI but did not require RRT.
Major Critiques
- The attempt at prediction modeling does not address commonly recognized causes of AKI including hemodynamic variables and medications. The cause of AKI is not described although presumed to be ATN, yet some information in this regard should be outlined.
- The most significant cohort by size is the patient group that had AKI but did not require RRT. What distinguishes this group which recovered without RRT from the patients that did require RRT?
- The tidal volume as a predictor requires some additional explanation. The mean tidal volume for the four populations in Table 1 is almost identical with no statistical difference. Yet tidal volume, outlined as per 1mg/kg increment, is a highly predictive variable in univariate and multivariate analysis. Is this latter variable a modification of tidal volume/ideal body weight? The differences in tidal volume must be small and this raises question about the clinical validity/relevance of this finding.
Minor Critiques
- Table 1 compares 4 diagnostic categories with a single p value reported. The presentation does not give a clear understanding of which diagnostic category is statistically significant.
- Table 1 does not include the AKI without RRT population which is a significant cohort and would be important to the overall analysis
- Table 1 does not include any hemodynamic data (e.g need for vasopressor agents)
- Table 2 is difficult to read and could be condensed without all the individual p values
- Table 3 requires some clarification in the methods section regarding how multiple admission day variables were chosen ie worst value / mean of values.
Author Response
Response to Reviewer 2 Comments
The investigators present a retrospective review of AKI requiring RRT in patients with influenza pneumonia from eight tertiary referral centers in Taiwan between Jan and Mar 2016.
The study cohort consisted of 282 patient admitted with influenza that met Berlin criteria for the diagnosis of ARDS. Sixteen patients were classified as ESRD and therefore would not be considered AKI patients. The analysis is based on the remaining 34 patients, of which, 10 med criteria for CKD prior to the hospitalization (GFR < 60). 76 patients met criteria for AKI but did not require RRT.
Major Critiques
- The attempt at prediction modeling does not address commonly recognized causes of AKI including hemodynamic variables and medications. The cause of AKI is not described although presumed to be ATN, yet some information in this regard should be outlined.
We thank the reviewer to point out this problem. There are several etiologies may induced acute kidney injury such as sepsis, volume depletion, medication or post-renal disease (1), and it is common that multiple factors induce AKI in critical patients. However, in our study, we focus on influenza pneumonia induced acute respiratory distress syndrome patients that the respiratory problem induced multi-organ failure or septic shock were the main reason of AKI that the patients with AKI requiring RRT had significant poorer severity score and PaO2/FiO2 ratio and they needed significantly more vasopressor agents. In the patients with AKI requiring RRT, the average period from ARDS onset to start RRT was 3.58±3.53 days that was shorter than the duration of nephrotoxic antibiotics induced AKI in previous study (12.1±9.6 days) (2), and no patients used obvious nephrotoxic antibiotics such as aminoglycoside or colistin before RRT was recorded. The Foley catheter placement was a routine management in ICU, and the post-renal etiology was not the cause of AKI. Moreover, in our study, we found that most patients had recovered from AKI and only one survival patient needed long term hemodialysis after the ARDS was improved, that was another evidence to imply the etiology of AKI. Furthermore, we did not do the renal biopsy routinely in critical care patients, so it was difficult to collect the pathologic change of kidney in these patients.
We added this sentence in 2.3 Renal replacement therapy in 2.Materials and Methods (page 3, line 123-125 in revised manuscript) as “Every patients had used Foley catheter to monitor the urine output and avoid post-renal etiology of renal failure. The nephrotoxic antibiotics such as aminoglycoside or colistin were not the first-line antibiotics in our ICUs.”
- Rahman M, Shad F, Smith MC. Acute kidney injury: a guide to diagnosis and management. Am Fam Physician. 2012;86(7):631-9.
- Khalili H, Bairami S, Kargar M. Antibiotics induced acute kidney injury: incidence, risk factors, onset time and outcome. Acta Med Iran. 2013;51(12):871-8.
- The most significant cohort by size is the patient group that had AKI but did not require RRT. What distinguishes this group which recovered without RRT from the patients that did require RRT?
We thank the reviewer to point out this problem. We added the table 2 to compare the patients with AKI requiring hemodialysis and not requiring hemodialysis as follows.
Table 2. Comparison of the patients with AKI hemodialysis and not requiring RRT
|
Characteristics |
AKI not requiring RRT |
AKI requiring RRT |
P value |
|
(n=76) |
(n=34) |
||
|
Age (years) |
61.8±16.7 |
55.8±12.4 |
0.064 |
|
Gender (male/female) |
47/29 |
24/10 |
0.376 |
|
BMI (kg/m2) |
24.3±5.1 |
26.0±5.0 |
0.111 |
|
Comorbidity |
|
|
|
|
Malignancy |
11 (14.5%) |
5 (14.7%) |
0.975 |
|
Liver disease |
12 (15.8%) |
10 (29.4%) |
0.099 |
|
Heart failure |
10 (13.2%) |
4 (11.8%) |
>0.999 |
|
Hypertension |
33 (43.4%) |
14 (41.2%) |
0.826 |
|
Diabetes Mellitus |
20 (26.3%) |
13 (38.2%) |
0.207 |
|
Chronic lung disease |
8 (10.5%) |
3 (8.8%) |
>0.999 |
|
Chronic steroid usage |
6 (7.9%) |
2 (5.9%) |
>0.999 |
|
Influenza type |
|
0.077 |
|
|
Type A |
50 (65.8%) |
29 (85.3%) |
|
|
Type B |
9 (11.8%) |
3 (8.8%) |
|
|
Unknown |
17 (22.4%) |
2 (5.9%) |
|
|
Severity index |
|
|
|
|
PSI |
120.6±45.4 |
142.0±46.2 |
0.025* |
|
CURB 65 |
2.5±1.3 |
2.4±1.0 |
0.571 |
|
APACHE II score |
23.4±7.4 |
29.5±8.1 |
<0.001* |
|
SOFA score |
10.3±3.8 |
13.9±3.5 |
<0.001* |
|
Laboratory data |
|
|
|
|
White blood cell (/uL) |
10075.8±6124.9 |
10810.6±7582.1 |
0.754 |
|
Platelet (1000/uL) |
156.7±75.8 |
136.4±85.1 |
0.113 |
|
C-reactive protein (mg/dL) |
14.9±10.3 |
19.4±10.2 |
0.041* |
|
Lactate (mg/dL) |
26.8±23.9 |
49.8±63.6 |
0.254 |
|
Albumin (mg/dL) |
2.9±0.5 |
2.7±0.6 |
0.060 |
|
Total bilirubin (mg/dL) |
0.9±0.8 |
1.7±2.0 |
0.369 |
|
Need vasopressor agents |
31 (40.8%) |
25 (73.5%) |
0.002* |
|
Arterial blood gas |
|
|
|
|
pH |
7.4±0.1 |
7.3±0.1 |
0.255 |
|
PaO2 (mm Hg) |
100.6±53.2 |
81.4±46.6 |
0.084 |
|
PaCO2 (mm Hg) |
43.3±15.9 |
42.3±18.6 |
0.781 |
|
HCO3 (mm/L) |
22.9±5.4 |
19.6±4.5 |
0.007 |
|
FiO2 |
0.8±0.2 |
0.9±0.2 |
0.035 |
|
PaO2/FiO2 ratio (mm Hg) |
105.7±62.0 |
82.4±46.7 |
0.037* |
|
Tidal volume/predict body weight (ml/kg) |
8.5±1.9 |
8.9±2.4 |
0.363 |
|
Positive end expiratory pressure (cm H2O) |
10.1±4.2 |
11.2±3.5 |
0.109 |
|
Peak airway pressure (cm H2O) |
29.1±4.7 |
31.6±5.1 |
0.028* |
|
Dynamic driving pressure (cm H2O) |
18.8±4.8 |
20.3±5.2 |
0.178 |
|
Compliance (mL/cm H2O) |
27.3±10.8 |
26.3±9.0 |
0.910 |
We add this paragraph in 3.1 General Data in 3.Results (page 4-5, line 178-186 in revised manuscript) as “Seventy-six patients were met the diagnosis of AKI, but they did not need RRT during the ICU course. The comparison of the AKI patients requiring RRT and not requiring RRT are shown in Table 2. The patients with AKI requiring RRT, comparing with the patients with AKI not requiring RRT, had significant higher PSI (142.0±46.2 vs. 120.6±45.4, p value=0.025), APACHE II score (29.5±8.1 vs. 23.4±7.4, p value<0.001), SOFA score (13.9±3.5 vs. 10.3±3.8, p value<0.001), C-reactive protein (19.4±10.2 vs. 14.9±10.3, p value=0.041), and peak airway pressure (31.6±5.1 vs. 29.1±4.7, p value=0.028), and significant poorer PaO2/FiO2 ratio (82.4±46.7 vs. 105.7±62.0, p value=0.037). More patients with AKI requiring RRT needed vasopressor agents than not requiring RRT (73.5% vs. 40.8%, p value=0.002)”
The sentence “There were 76 patients who had AKI without RRT during their ICU course, and their 60 day mortality rate was significantly lower than those patients who had AKI requiring RRT (30.3% vs. 58.8%, p value = 0.005).” in 3.1 General data in 3.Results (page 3, line 149-152 in revised manuscript) was moved to 3.2 Clinical outcomes in 3.Results (page 7, line 205-207 in revised manuscript) and was modified to “The patients who had AKI without RRT during their ICU course had significantly lower 60 day mortality rate than those patients who had AKI requiring RRT (30.3% vs. 58.8%, p value = 0.005).”
- The tidal volume as a predictor requires some additional explanation. The mean tidal volume for the four populations in Table 1 is almost identical with no statistical difference. Yet tidal volume, outlined as per 1mg/kg increment, is a highly predictive variable in univariate and multivariate analysis. Is this latter variable a modification of tidal volume/ideal body weight? The differences in tidal volume must be small and this raises question about the clinical validity/relevance of this finding.
We thank the reviewer to point out this problem. The tidal volume/predicted body weight had no significant difference between patients without renal replacement therapy, patients with AKI requiring RRT, and patients with ESRD. However, the tidal volume/predicted body weight was lower in survival patients compared to mortality patients in total patients (8.2±1.9 vs. 9.2±2.1, p value=0.001) and AKI requiring RRT patients (7.9±2.3 vs. 9.3±2.4, p value=0.077). Hence, the tidal volume/predicted body weight was a predictive variable in univariate and multivariate analysis for total patients (table 3) and AKI requiring RRT patients (table 5).
The lower tidal volume/predicted body weight correlated to lower mortality rate in ARDS patients was also demonstrated in another study by our group (3) or in randomized control studies (4). In our study, we found the lower tidal volume/predicted body weight was also an independent predictive factor for mortality in AKI requiring RRT patients.
- Chan MC, Chao WC, Liang SJ, Tseng CH, Wang HC, Chien YC, et al. First tidal volume greater than 8 mL/kg is associated with increased mortality in complicated influenza infection with acute respiratory distress syndrome. J Formos Med Assoc. 2019;118(1 Pt 2):378-85.
- Acute Respiratory Distress Syndrome N, Brower RG, Matthay MA, Morris A, Schoenfeld D, Thompson BT, et al. Ventilation with lower tidal volumes as compared with traditional tidal volumes for acute lung injury and the acute respiratory distress syndrome. N Engl J Med. 2000;342(18):1301-8.
Minor Critiques
- Table 1 compares 4 diagnostic categories with a single p value reported. The presentation does not give a clear understanding of which diagnostic category is statistically significant.
We thank the reviewer to point out this problem. The post hoc analysis was done to analyze the parameters with significant difference in ANOVA, and the result was shown in 3.1 General Data in 3.Results (Page 4, line 158-177 in revised manuscript) as follows.
The patients’ demographic data, laboratory data and ventilator parameters are shown in Table 1. The patients with AKI requiring RRT were younger than the no RRT patients and the ESRD patients, but these differences were not statistically significant. The patients with AKI requiring RRT had a higher ratio of underlying chronic liver disease or chronic kidney disease compared with the other patients, whereas the ESRD patients had a higher ratio of underlying diabetes mellitus and hypertension. The AKI patients requiring RRT and ESRD patients had significantly more severe conditions according to their PSI (AKI requiring RRT vs. no RRT: p value = 0.005, ESRD vs. no RRT: p value = 0.150), APACHE II (AKI requiring RRT vs. no RRT: p value = <0.001, ESRD vs. no RRT: p value = 0.002), and SOFA scores (AKI requiring RRT vs. no RRT: p value = <0.001, ESRD vs. no RRT: p value = 0.001), and they had significantly higher lactate (AKI requiring RRT vs. no RRT: p value = 0.003) and total bilirubin levels (AKI requiring RRT vs. no RRT: p value = 0.023) than no RRT patients.. The platelet level was lower in patients who received RRT, but no significant difference in post hoc analysis (AKI requiring RRT vs. no RRT: p value = 0.150, ESRD vs. no RRT: p value = 0.097). More severe metabolic acidosis was noted in the AKI patients requiring RRT, but this difference was not statistically significant. Moreover, the patients with AKI requiring RRT had a poorer PaO2/FiO2 ratio (AKI requiring RRT vs. no RRT: p value = 0.028), and they needed higher peak airway pressure (AKI requiring RRT vs. no RRT: p value = 0.020).
- Table 1 does not include the AKI without RRT population which is a significant cohort and would be important to the overall analysis
We thank the reviewer to point out this problem. To compare the AKI patients requiring or not requiring RRT, we added the table 2 as follows.
Table 2. Comparison of the patients with AKI hemodialysis and not requiring hemodialysis
|
Characteristics |
AKI not requiring RRT |
AKI requiring RRT |
P value |
|
(n=76) |
(n=34) |
||
|
Age (years) |
61.8±16.7 |
55.8±12.4 |
0.064 |
|
Gender (male/female) |
47/29 |
24/10 |
0.376 |
|
BMI (kg/m2) |
24.3±5.1 |
26.0±5.0 |
0.111 |
|
Comorbidity |
|
|
|
|
Malignancy |
11 (14.5%) |
5 (14.7%) |
0.975 |
|
Liver disease |
12 (15.8%) |
10 (29.4%) |
0.099 |
|
Heart failure |
10 (13.2%) |
4 (11.8%) |
>0.999 |
|
Hypertension |
33 (43.4%) |
14 (41.2%) |
0.826 |
|
Diabetes Mellitus |
20 (26.3%) |
13 (38.2%) |
0.207 |
|
Chronic lung disease |
8 (10.5%) |
3 (8.8%) |
>0.999 |
|
Chronic steroid usage |
6 (7.9%) |
2 (5.9%) |
>0.999 |
|
Influenza type |
|
0.077 |
|
|
Type A |
50 (65.8%) |
29 (85.3%) |
|
|
Type B |
9 (11.8%) |
3 (8.8%) |
|
|
Unknown |
17 (22.4%) |
2 (5.9%) |
|
|
Severity index |
|
|
|
|
PSI |
120.6±45.4 |
142.0±46.2 |
0.025* |
|
CURB 65 |
2.5±1.3 |
2.4±1.0 |
0.571 |
|
APACHE II score |
23.4±7.4 |
29.5±8.1 |
<0.001* |
|
SOFA score |
10.3±3.8 |
13.9±3.5 |
<0.001* |
|
Laboratory data |
|
|
|
|
White blood cell (/uL) |
10075.8±6124.9 |
10810.6±7582.1 |
0.754 |
|
Platelet (1000/uL) |
156.7±75.8 |
136.4±85.1 |
0.113 |
|
C-reactive protein (mg/dL) |
14.9±10.3 |
19.4±10.2 |
0.041* |
|
Lactate (mg/dL) |
26.8±23.9 |
49.8±63.6 |
0.254 |
|
Albumin (mg/dL) |
2.9±0.5 |
2.7±0.6 |
0.060 |
|
Total bilirubin (mg/dL) |
0.9±0.8 |
1.7±2.0 |
0.369 |
|
Need vasopressor agents |
31 (40.8%) |
25 (73.5%) |
0.002* |
|
Arterial blood gas |
|
|
|
|
pH |
7.4±0.1 |
7.3±0.1 |
0.255 |
|
PaO2 (mm Hg) |
100.6±53.2 |
81.4±46.6 |
0.084 |
|
PaCO2 (mm Hg) |
43.3±15.9 |
42.3±18.6 |
0.781 |
|
HCO3 (mm/L) |
22.9±5.4 |
19.6±4.5 |
0.007 |
|
FiO2 |
0.8±0.2 |
0.9±0.2 |
0.035 |
|
PaO2/FiO2 ratio (mm Hg) |
105.7±62.0 |
82.4±46.7 |
0.037* |
|
Tidal volume/predict body weight (ml/kg) |
8.5±1.9 |
8.9±2.4 |
0.363 |
|
Positive end expiratory pressure (cm H2O) |
10.1±4.2 |
11.2±3.5 |
0.109 |
|
Peak airway pressure (cm H2O) |
29.1±4.7 |
31.6±5.1 |
0.028* |
|
Dynamic driving pressure (cm H2O) |
18.8±4.8 |
20.3±5.2 |
0.178 |
|
Compliance (mL/cm H2O) |
27.3±10.8 |
26.3±9.0 |
0.910 |
We add this paragraph in 3.1 General Data in 3.Results (page 4-5, line 178-186 in revised manuscript) as “Seventy-six patients were met the diagnosis of AKI, but they did not need RRT during the treatment course. The comparison of the AKI patients requiring RRT and not requiring RRT are shown in Table 2. The patients with AKI requiring RRT, comparing with the patients with AKI not requiring RRT, had significant higher PSI (142.0±46.2 vs. 120.6±45.4, p value=0.025), APACHE II score (29.5±8.1 vs. 23.4±7.4, p value<0.001), SOFA score (13.9±3.5 vs. 10.3±3.8, p value<0.001), C-reactive protein (19.4±10.2 vs. 14.9±10.3, p value=0.041), and peak airway pressure (31.6±5.1 vs. 29.1±4.7, p value=0.028), and significant poorer PaO2/FiO2 ratio (82.4±46.7 vs. 105.7±62.0, p value=0.037). More patients with AKI requiring RRT needed vasopressor agents than not requiring RRT (73.5% vs. 40.8%, p value=0.002)”
- Table 1 does not include any hemodynamic data (e.g need for vasopressor agents)
We thank the reviewer to point out this problem. We added the data of “need for vasopressor agents” in the table 1 as follows.
|
Characteristics |
Total patients |
No RRT |
ESRD |
AKI requiring RRT |
P value |
|
Need vasopressor agents |
150 (53.2%) |
114 (49.1%) |
11 (68.8%) |
25 (73.5%) |
0.013* |
We also added the sentence “More patients with AKI requiring RRT needed vasopressor agents (AKI requiring RRT vs. no RRT: p value = 0.008, ESRD vs. no RRT: p value = 0.129).” in the paragraph 3.1 General data in 3.Result (Page 4, line 169-171 in revised manuscript).
- Table 2 is difficult to read and could be condensed without all the individual p values
We thank the reviewer to point out this problem. The table 2 was modified that the post hoc p value were deleted. The post hoc data was still presented in the paragraph 3.2 Clinical outcomes in 3.Result. The modified table 2 was as follows.
|
Characteristics |
No RRT |
ESRD |
AKI requiring RRT |
P value |
|
(n=232) |
(n=16) |
(n=34) |
||
|
Mortality – no. (%) |
||||
|
At day 30 |
46 (19.8%) |
2 (12.5%) |
17 (50.0%) |
<0.001* |
|
At day 60 |
63 (27.2%) |
5 (31.3%) |
20 (58.8%) |
0.001* |
|
Hospital |
70 (30.2%) |
5 (31.3%) |
20 (58.8%) |
0.004* |
|
Length of ICU stay |
||||
|
Survivors |
19.8±18.5 |
25.6±22.0 |
29.9±23.3 |
0.125 |
|
Non-survivors |
20.1±18.2 |
15.6±12.5 |
13.3±10.5 |
0.298 |
|
Length of hospital stay |
||||
|
Survivors |
35.3±25.9 |
46.8±35.1 |
58.8±42.9 |
0.007* |
|
Non-survivors |
26.3±24.6 |
33.8±20.3 |
16.8±10.9 |
0.162 |
|
Ventilation-free days |
||||
|
At day 30 |
11.5±9.8 |
10.1±10.4 |
4.1±7.6 |
<0.001* |
|
At day 60 |
31.3±22.3 |
28.4±22.1 |
13.8±19.4 |
<0.001* |
- Table 3 requires some clarification in the methods section regarding how multiple admission day variables were chosen ie worst value / mean of values.
We thank the reviewer to point out this problem and we apologized for the negligence. We added this sentence in 2.4 Statistical analyses in 2.Materials and Methods (Page 3, line 137-138 in revised manuscript) as “Variables with p value less than 0.05 in univariate analysis were included for multivariate analysis.”.
Please see the attachment.

Round 2
Reviewer 2 Report
Thank you. The authors have addressed the critiques within the limitations of the data.
This manuscript is a resubmission of an earlier submission. The following is a list of the peer review reports and author responses from that submission.
Round 1
Reviewer 1 Report
Albeit retrospective, this study is of interest to the general ICU medical public. The topic is one that is poorly investigated so far. the methodology is sound, the conclusions are clear, the statistics are clearly displayed and the discussion is right to the topic. I think the introduction may be shortened.
Reviewer 2 Report
1) General comments
In the present study, Chang and coworkers evaluated risk factors and outcomes related to renal replacement therapy (RRT) in patients with influenza pneumonia-related acute respiratory distress syndrome. It is a retrospective cohort study that included 336 patients admitted to 8 ICUs in Taiwan by obtaining data from electronic medical records. The authors found that the 30- and 60-day mortality rates were significantly higher in patients with acute kidney injury (AKI) requiring RRT compared with those not requiring RRT, but the patients with end-stage renal disease (ESRD) had no significant difference in mortality compared with those not requiring RRT. The number of patients in the group of AKI requiring RRT and ESRD was 34 and 16, respectively.
2) Specific comments:
- a) Major concerns
- Driving pressure that is calculated as plateau pressure minus positive end-expiratory pressure (PEEP) is generally used to consider ventilatory settings for lung-protective ventilation. The authors have to explain why they used dynamic driving pressure defined as peak inspiratory pressure minus PEEP instead of driving pressure.
- There are no statements that the authors did power analysis. The authors have to confirm that the number of patients they included and analyzed is enough to draw a conclusion without either type 1 or type 2 errors. In a retrospective study like the present study, the authors might have chosen a study period by predicting the number of patients distributed to each group by referring to the previous demography of patients of each study site.
- Without a power analysis, some of the conclusions that the authors have made are not appropriate. For example, they concluded that the patients with ESRD on regular hemodialysis did not have a significantly different mortality rate compared with the patients without RRT and discussed that the result does not match the results in previous papers (refs. #24 to #26); however, those papers included a large number of patients, about 280000, 39000, and 160000, respectively.
- Analysis of variance cannot find differences between groups unless doing post hoc analysis (p. 4 L. 142 to 153).
- b) Minor concerns
- There are discrepancies between numbers in the text and those listed in Tables (p-value of 0.473 in p. 5, L. 164 vs. 0.744 in Table 2 as an example).
- Explanations of axes in Figure 2 should be corrected: Y-axis should be survival and X-axis should be death.
Reviewer 3 Report
Reviewer comments on the manuscript: „Renal Replacement Therapy in Patients with Influenza Pneumonia Related Acute Respiratory Distress Syndrome”
- General comments
The manuscript is very well written. The authors report interesting data n an interesting and clinically most relevant issue. However, serious concerns arose with respect to the fact that major conclusions may not be supported by the data presented (see major comments)
- Major comments
- The authors should explain clearly their major finding that RRT in AKI is a predictor for death in this cohort. Instead, it seems as if the subgroup of patients that required RRT were generally sicker – most of the clinically relevant baseline parameters were worse in this cohort than in the non-RRT cohort. Therefore, it seems more likely that higher death in this subgroup is driven by the overall disease status and not specifically by the renal failure. This is critical for the overall interpretation of the main results.
- Minor comments
- The IRB approval identifyer should be mentioned.
- Design of figure 1 should be revised so that all lines are displayed correctly.
